# OpenReview forum: "Mechanistic Unlearning: Robust Knowledge Unlearning and Editing via Mechanistic Localization"
_ICML.cc/2025/Conference — ICML 2025 spotlightposter_

### Official Review · Reviewer_wVvy · 2025-03-02

**Overall Recommendation:** 2

**Summary:**

The author investigates how mechanistic interpretability improves the precision and robustness of knowledge editing and unlearning in LLMs. They distinguish between methods that preserve outputs and those that target high-level mechanisms with predictable states. The findings show that localizing edits to lookup-table mechanisms for factual recall enhances robustness across formats, resists relearning attacks, and reduces unintended side effects, outperforming baselines on the sports facts and CounterFact datasets. Additionally, certain localized edits disrupt latent knowledge more effectively, making unlearning more resilient to adversarial attacks.

**Claims And Evidence:**

Not clear. Please see Question For Authors.

**Essential References Not Discussed:**

None

**Experimental Designs Or Analyses:**

I think this paper's experimental setup has issues. Please refer to Questions for Authors for details.

**Methods And Evaluation Criteria:**

I think this paper's experimental setup has issues. Please refer to Questions for Authors for details.

**Other Comments Or Suggestions:**

Please refer to Questions for Authors for details.

**Other Strengths And Weaknesses:**

Please refer to Questions for Authors for details.

**Questions For Authors:**

1. The authors repeatedly mention machine unlearning but do not adopt any well-known machine unlearning methods in their paper, such as RMU [1], GradDiff [2], or NPO [3]. Based on the presented results, it is unclear whether their proposed approach truly outperforms existing machine unlearning methods.

> [1] Li, Nathaniel, et al. "The WMDP Benchmark: Measuring and Reducing Malicious Use with Unlearning." arXiv preprint arXiv:2403.03218 (2024).

> [2] Yao, Yuanshun, Xiaojun Xu, and Yang Liu. "Large Language Model Unlearning." Advances in Neural Information Processing Systems 37 (2025): 105425-105475.

> [3] Zhang, Ruiqi, et al. "Negative Preference Optimization: From Catastrophic Collapse to Effective Unlearning." arXiv preprint arXiv:2404.05868 (2024).

2. Additionally, the authors do not conduct experiments on any established unlearning benchmarks, such as WMDP [1], TOFU [2], or MUSE [3].

> [1] Li, Nathaniel, et al. "The WMDP Benchmark: Measuring and Reducing Malicious Use with Unlearning." arXiv preprint arXiv:2403.03218 (2024).

> [2] Maini, Pratyush, et al. "TOFU: A Task of Fictitious Unlearning for LLMs." arXiv preprint arXiv:2401.06121 (2024).

> [3] Shi, Weijia, et al. "MUSE: Machine Unlearning Six-Way Evaluation for Language Models." arXiv preprint arXiv:2407.06460 (2024).

3. The proposed method merely applies existing model editing techniques to the field of machine unlearning without introducing any novel contributions.

4. The authors claim that their approach provides robust unlearning, yet they only evaluate it against a single technique—relearning attacks. What about other methods? For instance, adversarial prompts, logit lens, etc. [1].

> [1] Łucki, Jakub, et al. "An Adversarial Perspective on Machine Unlearning for AI Safety." arXiv preprint arXiv:2409.18025 (2024).


## Update after rebuttal
This paper does not compare its method against well-known machine unlearning benchmarks such as TOFU, MUSE, or WMDP, nor does it evaluate against established unlearning methods like NPO or RMU. As a result, it is difficult to assess the effectiveness of the proposed approach in the context of machine unlearning. The authors' rebuttal does not fully address my concerns. But I appreciate their efforts and am willing to raise my score to 2.

**Relation To Broader Scientific Literature:**

Please refer to Questions for Authors for details.

**Theoretical Claims:**

There is no theoretical claims.

---

> ### Author Rebuttal · Authors · 2025-04-01
>
> Thank you for reading our work and providing feedback. We appreciate the opportunity to clarify our contributions and address your concerns. We believe that the assessment and low score was based on certain misunderstandings of our work: we focus on model editing of factual relations; our editing method is in fact novel and clears up misconceptions about mechanistic interpretability for model editing and unlearning, making it an important contribution to the literature; we provide an extensive set of evaluations of our method, and even stronger attacks than suggested by the reviewer.
>
> We hope our detailed responses below will convince the reviewer to re-evaluate our work and raise the score.
>
> *On the Lack of Unlearning benchmarks*
>
> Our primary contribution lies in investigating how mechanistic interpretability can enhance the precision and robustness of knowledge editing of factual associations: i.e., replacement of certain facts with new facts. This goal underscores our choices of baselines and datasets. We use the SportsFacts dataset following work from Nanda et. al, who use it to mechanistically understand factual recall, to create a localization technique for robustly modifying factual associations. We then translate these findings to the CounterFact dataset, a benchmark widely used in the editing literature.
>
> Most of our work focuses on editing rather than unlearning, and the baselines of RMU, GradDiff, and NPO couldn’t be used for the editing objectives without a significant reformulation of the tasks. Our inclusion of an unlearning result in A.1 of the paper was primarily to show the potential for our method to generalize to unlearning, but not a claim that our method led to state of the art unlearning performance on every benchmark. However, the current literature on unlearning is relatively unanimous in that no current method is robust against the partial relearning attack (Deeb 2024), including RMU (Li 2024) and TAR (Tamirisa 2024) which is defeated by parameter-efficient fine-tuning: we believe our method can yield progress.
>
> In this work, we hoped to lay the groundwork for applying interpretability for unlearning. We are excited about future work that applies more sophisticated interpretability on complex datasets like WMDP: a positive outcome of the results of this paper would be to inspire further research into interpretability to achieve more robust editing and unlearning.
>
> We will revise the paper to more accurately reflect that the majority of our current empirical results focus on fact editing. An easy fix!
>
> *On Novelty*
>
> One main novel contribution is the identification and utilization of Fact Lookup (FLU) mechanisms for robust knowledge editing. We contrast this approach with Output Tracing (OT) methods. We show that editing localized to these FLU mechanisms leads to more robust edits as measured using a number of evaluations (listed below).
>
> We demonstrate that the relationship between localization and fact editing/unlearning is more nuanced than suggested in Hase et al. (2023),  and that not all localization techniques are equally effective. This is an important contribution to the literature, showing a promising use of mechanistic interpretability.
>
> *On evaluations*
>
> Regarding the evaluation against different attacks, we would like to highlight that we do evaluate the robustness of our method against:
> * Rephrasing prompts (Paraphrase Evaluation);
> * Multiple-choice question extraction (MCQ Evaluation);
> * Adversarial relearning attacks;
> * Soft prompt attacks, which are a more challenging form of adversarial prompting since they operate in continuous space.
> * Latent knowledge analysis, which trains a probe to extract the correct answer from the latent representation of the model, instead of logit lens which is a biased representation of the model’s best guess at a layer
>
> We are confident that these evaluations provide a comprehensive assessment of the robustness of our proposed editing method.
>
> Note that the current version of the paper is even stronger as we included additional experiments suggested by other reviewers: proposed and tested an automated version of our localization method which outperforms strong baselines; included additional tests on which components are important for localization; and demonstrated that our method can successfully edit a large number of facts (up to 1000).
>
> References:
> Hase, P. et al. Does localization inform editing? surprising differences in causality-based localization vs. knowledge editing in language models, 2023.
>
> Meng, K. et al. Locating and editing factual associations in gpt, 2023.
>
> Nanda, N. Attribution patching: Activation patching at industrial scale, 2023.
>
> Li, N. et al. The WMDP Benchmark: Measuring and Reducing Malicious Use With Unlearning, 2024.​
>
> Deeb, A. et al. Do Unlearning Methods Remove Information from Language Model Weights?, 2024.​
>
> Tamirisa, R. et al. Tamper-Resistant Safeguards for Open-Weight LLMs, 2024.​

---

### Official Review · Reviewer_MshQ · 2025-03-14

**Overall Recommendation:** 3

**Summary:**

The authors investigate the effectiveness of adopting techniques from mechanistic interpretability to improve editing and unlearning in large language models. In particular, the work focuses on analyzing the benefits of unlearning and editing brought by localization techniques based on factual lookup (FLU) instead of the typical strategies using causal tracing methods. The authors focus their experiments on a sports fact dataset and a counterfactual dataset, showing that, in these contexts, their methodology leads to robust unlearning/editing while mitigating the risks of relearning attacks.

**Claims And Evidence:**

The idea of studying editing and unlearning from a mechanistic interpretability perspective is interesting and timely, as well as the approach of focusing on the role of the factual recall mechanism in editing and unlearning of LLMs knowledge.
The authors consider diverse models to show that their findings are robust to medium-level LLM scale and to different pre-training strategies (e.g., dataset, hyperparameters, etc.). Some points could make the analysis more convincing (see Section "Other Strengths And Weaknesses").

**Essential References Not Discussed:**

NA

**Experimental Designs Or Analyses:**

The experimental design is well constructed and focused on supporting the main claims. Some extensions might strengthen the work (see Section "Other Strengths And Weaknesses")

**Methods And Evaluation Criteria:**

The evaluation of the unlearning and editing performances on the sports fact dataset and CounterFact dataset is sound and clearly explained. Some extensions of the experimental setup would strengthen the evidence for the claims (see Section "Other Strengths And Weaknesses").

**Other Comments Or Suggestions:**

I have no further comments or suggestions.

**Other Strengths And Weaknesses:**

1) It would be beneficial to justify better the choice of focusing only on interventions on MLP layers. Even if, as discussed by the author, they play a crucial role in the factual recall process and thus are a natural candidate for editing and unlearning, showing that interventions that involve the attention mechanism do not allow to achieve better performance would give stronger evidence for the author's choices.

2) It would be beneficial to at least outline a clear semi-automatic procedure for the selection of model components, even if the current work is more focused on giving proof of concept. In particular, is it possible to deduce a strategy from the "manual analysis for both datasets outlined in Appendix A.2.1"?

3) It would be beneficial to extend the experiments to at least one additional more challenging scenario. For instance a subset of subjects in MMLU could provide an example.

4) In the reviewer's opinion, even if averaging over models allows for simplifying the presentation, it would be important to show error bars to show that the trends are shared among models or to find other strategies to support this (e.g., separate results in the appendix).

**Questions For Authors:**

I have no further questions.

**Relation To Broader Scientific Literature:**

The authors discuss at a good level of detail the relevant literature for the current submission.

**Theoretical Claims:**

NA

---

> ### Author Rebuttal · Authors · 2025-04-01
>
> Thank you for your detailed review and valuable proposals. In response, we proposed a way to automate our method, which allowed us to test it at scale in terms of the number of facts to be edited. We also added experiments demonstrating that adding attention heads does not improve performance.
>
> Addressing points 1, 2:
>
> We originally ignored the attention heads as candidates for localization as a result of work done by Nanda et. al who identified attention heads as playing a “fact extraction” role in the recall mechanism. We don’t claim that we found the precisely optimal localization for editing, but rather that this somewhat coarse localization was sufficient enough to yield significant robustness improvements: future work with more sophisticated interpretability techniques could further strengthen these results. However, we agree that empirical evidence to prove this would be valuable.
>
> To address this, along with your second point for a more automated strategy for localization and your third point for a more challenging editing scenario, we run an experiment scaling up the number of CounterFact facts edited from 64 to 1000, and try a new localization. We stick to CounterFact as the dataset contains factual questions in varying formats. Our difficulty increase thus comes from scaling the number of facts edited: our technique now has to maintain robustness to orders of magnitude greater number of facts. We do this experiment for the Gemma-7B model, and plan to replicate this setup across Gemma-2-9B and Llama-3-8B by the camera-ready deadline.
>
> This scaling of facts also means we have to use a more automated technique for localization. We localize per-fact, employing the same technique as described in A.2.2. We pick the components by utilizing a heuristic, picking all important MLPs that affect the final logit difference by > 2 stds. This differs from our original manual localization where we analyzed a group of facts, took the average contribution for each MLP, and assigned a group of MLPs to be the localization.
>
> We test our technique, based on your suggestion, to also include relevant attention heads. We compare this against a strong baseline of picking all the MLPs. We measure the robustness of these techniques to prompt changes (using an MCQ prompt format as in section 3.1) and to latent attacks by measuring probe accuracies, as in Section 3.3.
>
> We see that our fact lookup localization technique (localizing MLP layers only) maintains its MCQ forget error as we scale the facts edited, outperforming the baselines (fig: https://imgur.com/a/60azzEo). Our localization also outperforms other methods on its MCQ edit accuracy (fig: https://imgur.com/a/L7TnBwu), being the only localization to generalize to an alternative prompt setting. Finally, we see that latent knowledge attacks continue to fail when using our localization, as the probe accuracies are as accurate as random chance (fig: https://imgur.com/a/Fn6xWp1). All figures are anonymized and do not contain author information.
>
> Addressing point 3:
> In this work, we hoped to lay the groundwork for applying interpretability for unlearning. We chose to work with CounterFact and Sports Facts datasets because the mechanisms of factual recall in these datasets have been well studied in the literature, and because CounterFact is a benchmark widely used in the editing literature, including in the seminal paper by Meng et. al. We are excited about future work that applies more sophisticated interpretability analyses on complex datasets like MMLU: a positive outcome of the results of this paper would be to inspire further research into interpretability to achieve more robust editing and unlearning.
>
> Addressing point 4:
> We will add error bars and also results for each model across the various evaluations in the camera-ready version. For latent knowledge we do present results for each model in Appendix A.7.4 since each model has a different number of layers. For the latent knowledge analysis, Fact Lookup localization is most robust in Gemma-7b and Llama-3-8b, while some other methods are competitive with Fact Lookup in Gemma-2-9b.
>
> Thank you again for your feedback. We hope that these strong additional empirical results, automation of our method, and our clarifications addressed all of your concerns.
>
> References:
> [1] Nanda, N., Rajamanoharan, S., Kram  ́ar, J., and Shah, R. Fact finding: Attempting to reverse engineer factual recall on the neuron level, Dec 2023. URL https://www.alignmentforum.org/posts/iGuwZTHWb6DFY3sKB/fact-finding-attempting-to-reverse-engineer-factual-recall.

---

### Official Review · Reviewer_GkhG · 2025-03-17

**Overall Recommendation:** 3

**Summary:**

The paper studies the mechanistic localizations for knowledge unlearning and editing. There are two main categories of mechanistic localizations in the literature: Output Tracing and Fact Lookup. Through a designed experiment, the paper finds that Fact Lookup localizations make knowledge unlearning / editing more robust, at the aspect of rephrasing prompt, multi-choice question extraction and adversarial relearning.

**Claims And Evidence:**

The main conclusion is that localizing edits/unlearning to components associated with the
lookup-table mechanism is more robust, and this conclusion is well supported by the empirical results.

**Essential References Not Discussed:**

N/A

**Experimental Designs Or Analyses:**

The metrics and baselines are sufficient. The experiment is well aligned with the main question to check in this paper. However, I have some concerns for the dataset set-up. The dataset for the evaluation is quite small -- both two datasets only have 16 or 64 facts for editing, potentially making the empirical results sensitive to this specific test set. It is also unknown for the behaviors of editing larger batch of facts.

**Methods And Evaluation Criteria:**

Overall the evaluation set-up make sense. However, the description for one of the method remains unclear to me. Lines 171-196 describe the Fact Lookup localization for CounterFact, but the steps of the method are not clear after reading this paragraph. It might be necessary to more formally introduce these evaluated methods.

**Other Comments Or Suggestions:**

Figure 2 and Figure 4 are not mentioned / described in the main text.

**Other Strengths And Weaknesses:**

The paper is well written. I mostly enjoyed reading the paper and the presentation of the results.

**Questions For Authors:**

N/A

**Relation To Broader Scientific Literature:**

N/A

**Theoretical Claims:**

N/A

---

> ### Author Rebuttal · Authors · 2025-04-01
>
> Thank you for your comments, we will make sure to improve the description of the FLU localization technique for CounterFact in Sec 2.2, and we report new positive experimental results editing significantly more facts. The methodology is briefly summarized here:
>
> An important pre-requisite is “path patching”, described in section 3.1 of Wang et. al. This allows us to measure the importance of the direct edge between two components in a model. We first measure the direct effect of each attention head on the final output, using the logit difference between the original and edit answer as our measure. Nanda et. al show that these attention heads “extract” relevant associations from the residual stream. We call these the fact extraction heads.
>
> We then measure the effect of each MLP on the final logit difference as mediated only through these fact extraction heads. That is, we iterate through the MLPs and path patch the MLP -> {all fact extraction head} edges, and measure the logit difference. MLPs that cause a large change in this measure do so by introducing the factual association into the residual stream via a lookup mechanism, enabling the fact extraction heads to parse the association. We call these our fact lookup MLPs, which are the localization.
>
>
> Below we report new experimental results in response to your concern about the number of facts we edit.
>
> We acknowledge your point about the relatively small size of the dataset for editing. Our reasoning for maintaining a smaller set of facts was primarily to facilitate the creation of precise manual mechanistic localizations. However, we recognize the importance of understanding how our findings might generalize to a larger set of edited facts.
>
> To address this, we slightly modify our technique to localize per fact rather than averaging over a set of facts. Then, similar to our original technique in A.2.2, we pick the relevant components by identifying the MLPs that have a >2 std impact on the logit difference in the fact. Using this, we can precisely localize each fact at scale. This allowed us to run an additional evaluation scaling up the number of facts on the CounterFact dataset to be edited all the way to 1000 facts.
>
> We do this evaluation only on our Gemma-7b model due to time constraints, but we can expand this to Gemma-2-9B and Llama-3-8B as well by the camera-ready deadline. We compare our localization technique to using all the MLPs as a localization (a strong baseline) and a localization that uses our technique but also includes attention heads (as suggested by reviewer MshQ). We measure the robustness of these techniques to prompt changes (using an MCQ prompt format as in section 3.1) and to latent attacks by measuring probe accuracies as in section 3.3.
>
> We see that our fact lookup localization technique maintains its MCQ forget error as we scale the facts edited, outperforming the baselines (fig: https://imgur.com/a/60azzEo). Our localization also outperforms on its MCQ edit accuracy (fig: https://imgur.com/a/L7TnBwu), being the only localization to generalize to an alternative prompt setting. Finally, we see that latent knowledge attacks continue to fail when using our localization, as the probe accuracies are as accurate as random chance (fig: https://imgur.com/a/Fn6xWp1). All figures are anonymized and do not contain author information.
>
> We hope our additional experiments have convinced you of the robustness of our method at scale.
>
> References:
>
> Nanda, N., Rajamanoharan, S., Kram  ́ar, J., and Shah, R. Fact finding: Attempting to reverse engineer factual recall on the neuron level, Dec 2023. URL https://www.alignmentforum.org/posts/iGuwZTHWb6DFY3sKB/fact-finding-attempting-to-reverse-engineer-factual-recall.
>
> Wang, Kevin, et al. "Interpretability in the wild: a circuit for indirect object identification in gpt-2 small." arXiv preprint arXiv:2211.00593 (2022).

---

### Official Review · Reviewer_EZb3 · 2025-03-17

**Overall Recommendation:** 4

**Summary:**

This paper focuses on machine unlearning i.e. when the model needs to be prevented from outputing a certain information such as the profession of a person, or any mention of a given sport. Specifically they show that a lot of known methods might be preventing access to specific facts, but not overwriting the facts themselves. They show that their method on the other hand does this better.

**Claims And Evidence:**

Claim 1: Output tracing does not unlearn fact, it unlearns access to the fact which can be regained by different forms of relearning (Both evidence from the litterature and empirical examples are provided)
Claim 2: Fact lookup localisation is more likely to delete the fact itself. It is also more parameter efficient (This is verified through many ablations, and multi-dimensional evaluation. They also use probes to verify data presence)

**Essential References Not Discussed:**

To my knowledge, no key papers are missing

**Experimental Designs Or Analyses:**

Experimental design is sound. As previously stated, relevance of method is checked through different datasets, different levels of unlearning, different datasets, different parameters, giving a good overview that supports the generality of the claims

**Methods And Evaluation Criteria:**

They test there claims on 3 state of the art models of around 8B parameters, from two different companies.
They use different datasets,
They evaluate different state of the art methods, and perform extensive ablation

**Other Comments Or Suggestions:**

No additional comment

**Other Strengths And Weaknesses:**

Strong and well explained experimental process, relying on many tools which are effectively used and described. Benchmark is quite extensive, on multiple models, datasets, and different tasks. Variations and relevance of tasks are discussed. I particularly appreciate the attention to the different forgetting mechanisms, and the different possible components to consider.

**Questions For Authors:**

In 3.2 you mention that models should be able to generalize relearning half the basketball athletes to all basketball athletes. Could you further comment on how this informs the unlearning procedure? Are all athletes still known to perform in the same sport, and is basketball simply missing?

**Relation To Broader Scientific Literature:**

Relevant methods are discussed and explained, main drawbacks of said methods are carefully explained.

**Theoretical Claims:**

This paper is more on the empirical side. Experiments are nonetheless well defined and situated within the relevant litterature

---

> ### Author Rebuttal · Authors · 2025-04-01
>
> Thank you for a thorough read of our submission. Below we answer your question regarding the generalization of relearning in the Sports-Athlete-Editing and Full-Sports-Editing tasks, and explain our rationale behind the task choice for relearning attacks.
>
> Our relearning experiments in Section 3.2 are performed on the Sports-Athlete-Editing task. In this task, for a randomly selected set of athletes, we edit their associated sport by assigning them a new sport chosen uniformly at random from the existing set of sports.
>
> In this Sport-Athlete-Editing task, we do not expect relearning to “generalize”: there is no inherent correlation between an athlete, their original sport, and the newly assigned sport. Therefore, relearning a subset of edited facts should not provide the model with a basis to correctly infer the previously unlearned facts for other athletes, even if they were originally associated with the same sport like basketball (unless this old information is still stored in the model after editing).
>
> On the other hand, relearning attack is uninformative in some other setups, where the forget set accuracy could go up by the model simply learning a fixed mapping (e.g., always respond with “Golf” instead of “Basketball”). For example, this concern would apply for relearning the Full-Sports-Editing task: here we reassign one sport to another for all athletes playing that sport. For example, we might change all associations of "basketball" to "golf." If we then retrain the model on half of these edited facts, it's highly probable that the model would simply learn the global reassignment (e.g., all "golf" becomes "basketball") rather than truly relearning the specific original associations. This makes it difficult to distinguish between a newly learned association and the relearning of previously stored knowledge.
>
> It is important to note that the relearning paper we base our evaluations off (Deeb 2024) also makes efforts to avoid retraining on a set which does not have independent information from the rest of the forget set: when the information used for retraining is independent from the rest of the forget set, we don’t expect any information recovery given perfect unlearning/editing. However, when the retrained information is not independent from the rest of the forgotten facts, it is unclear what baseline amount of recovery is expected.
>
> We hope this clarifies our approach and the rationale behind our experimental design. We would also like to highlight that we ran additional experiments requested by other reviewers to further strengthen the paper: we showed that the localization part of our method can be automated, and that our method can be scaled to successfully edit a large number of facts.

---

### Decision · Program_Chairs · 2025-05-01

**Decision:**

Accept (spotlight poster)

**Comment:**

## Summary

This paper investigates how mechanistic interpretability can improve knowledge editing and unlearning in large language models. The authors distinguish between methods that localize components based primarily on preserving outputs and those finding high-level mechanisms with predictable intermediate states. They demonstrate that localizing edits/unlearning to components associated with the lookup-table mechanism for factual recall leads to more robust edits/unlearning across different input/output formats, better resistance to relearning attempts, and reduced unintended side effects. Experiments across multiple models on both sports facts and CounterFact datasets show that their localized edits disrupt latent knowledge more effectively than baselines, making unlearning more robust against various attacks.

## Reasons to Accept

1. The paper demonstrates that fact lookup (FLU) localization leads to more robust edits/unlearning across different input/output formats and resists attempts to relearn unwanted information, while reducing unintended side effects (EZb3, MshQ).
2. The experimental evaluation is comprehensive, with testing on multiple models (around 8B parameters from two different companies), multiple datasets, and extensive ablation studies (EZb3).
3. The research question combining mechanistic interpretability with model editing is interesting and timely, providing a novel perspective on how to improve edit robustness (MshQ, GkhG).
4. The main conclusion that localizing edits/unlearning to components associated with the lookup-table mechanism is more robust is well supported by the empirical results (GkhG).
5. The paper provides a strong and well-explained experimental process, with benchmarks that cover multiple models, datasets, and tasks (EZb3).

## Suggested Revisions

1. The scale of evaluation datasets is quite small (16 or 64 facts for editing), which makes the empirical results potentially sensitive to the specific test set and doesn't demonstrate behavior when editing larger batches of facts (GkhG). (The authors responded with new results scaling up to 1000 facts, showing their method maintains robustness at scale.)
2. The paper focuses only on interventions in MLP layers without justifying why interventions involving the attention mechanism are excluded (MshQ). (The authors ran new experiments demonstrating that adding attention heads does not improve performance, explaining that their focus on MLPs stems from previous work by Nanda et al.)
3. The paper lacks a clear semi-automatic procedure for the selection of model components, relying instead on manual analysis (MshQ). (The authors proposed an automated technique for localization in their response, demonstrating it works at scale.)

There are also some minor points:
1. The paper presents average results across models without error bars or separate results to show the trends are shared among different models (MshQ).
2. The description of the Fact Lookup localization method for CounterFact is unclear, and the steps of the method are not well-defined after reading this section (GkhG). Figures 2 and 4 are not mentioned or described in the main text (GkhG).